# Crossing Bacterial Genomic Features and Methylation Patterns with MeStudio: An Epigenomic Analysis Tool

**DOI:** 10.3390/ijms24010159

**Published:** 2022-12-21

**Authors:** Christopher Riccardi, Iacopo Passeri, Lisa Cangioli, Camilla Fagorzi, Marco Fondi, Alessio Mengoni

**Affiliations:** Department of Biology, University of Florence, Via Madonna del Piano 6, 50019 Sesto Fiorentino, Italy

**Keywords:** DNA methylation, mapping, genome sequencing, epigenomics

## Abstract

DNA methylation is one of the most observed epigenetic modifications. It is present in eukaryotes and prokaryotes and is related to several biological phenomena, including gene flow and adaptation to environmental conditions. The widespread use of third-generation sequencing technologies allows direct and easy detection of genome-wide methylation profiles, offering increasing opportunities to understand and exploit the epigenomic landscape of individuals and populations. Here, we present a pipeline named MeStudio, with the aim of analyzing and combining genome-wide methylation profiles with genomic features. Outputs report the presence of DNA methylation in coding sequences (CDSs) and noncoding sequences, including both intergenic sequences and sequences upstream of the CDS. We apply this novel tool, showing the usage and performance of MeStudio, on a set of single-molecule real-time sequencing outputs from strains of the bacterial species *Sinorhizobium meliloti*.

## 1. Introduction

Understanding organism adaptation to variable environmental conditions is pivotal for weighting the relevance of natural selection over species and population evolution. Phenotypic plasticity, stress responses, and acclimation contribute significantly to epigenetic mechanisms [1]. Among epigenetic modifications, DNA methylation has been shown to be essential in the control of several biological phenomena in eukaryotes and prokaryotes [2], and, in recent years, the study of variation in epigenetic response aroused the attention of several investigators [3]. Third-generation sequencing technologies, namely, single molecule real-time (SMRT) [4,5] and nanopore ONT [6,7] sequencing, allow rapidly identifying the most commonly methylated bases [8,9,10]. These methods are improving genome-wide DNA methylation studies, especially in prokaryotes, where the compact size of genomes allows the generation of whole-genome methylomes with relative ease. In prokaryotic microorganisms, DNA methylation plays various roles, which span from control of the cell cycle to protection against phages (e.g., restriction-modification systems) and regulation of gene expression (see, e.g., [11]). Relative to cell cycle control, genome-wide DNA methylation profiles have been shown to vary in ecologically relevant contexts (e.g., bacterial differentiation [12]), as well as for restriction-modification systems with respect to strain or population variation [12].

Consequently, the interest in computational pipelines which can easily profile DNA methylation features in a genome-wide manner (thus allowing a comparison of strains and individuals across multiple conditions) is growing. Several tools have been developed for the analysis of DNA methylation profiles deriving from bisulfite sequencing and microarrays (e.g., [13,14,15,16,17]; for a recent benchmarking, see [18]). Recently, three packages have been released [19,20,21], which are used to visualize methylation profiles from ONT sequencing data. A recent tool on GitHub was also developed to specifically analyze DNA methylation profiles on metagenomic data (https://github.com/hoonjeseong/Meta-epigenomics (accessed on 7 September 2022)). However, to the best of our knowledge, no specific pipeline has been developed for extracting DNA methylation information from sequencing data to allow a direct quantification/comparison of the position of methylated sites with respect to genome-derived features, such as coding and noncoding sequences and report outputs, which can be used in population epigenomic analyses. The position of methylated sites with respect to genomic features is of key importance in studies focusing on the role that epigenetic modifications have in gene expression control and phenotypic plasticity.

Here, we report the implementation of a bioinformatic tool, named MeStudio, to explore the methylation profiles and map the methylation patterns to genomic features on a set of genome sequences obtained by SMRT technology of the model symbiotic nitrogen-fixing bacterium *Sinorhizobium meliloti* [22] for which DNA methylation plays a relevant role in cell cycle regulation and differentiation during symbiotic conditions [23]. MeStudio is a pipeline for SMRT sequencing methylation data integration and visualization, combining methylation data with genome sequence and annotation to facilitate the extraction of biological information from DNA methylation profiles. Visual and tabular outputs are produced, which can be further processed to provide biological interpretation and formulate hypotheses on epigenomic profiles.

## 2. Results and Discussion

### 2.1. Tool-Wide Comparison

MeStudio provides a novel amount of feature-level information that is not present in other widely used genomic software packages. For instance, Bedtools (https://bedtools.readthedocs.io/en/latest/ (accessed on 17 August 2022)) is a well-known toolset for genomic applications through which it is possible to detect methylation features regarding CpG island, but it is not possible to extract information about CDS, nCDS, tIG, and US regions as it does not provide any figure about methylated motif occurrences. Bioconductor also supplies packages that can be used for methylation analysis such as “*GenomicRanges*” (https://bioconductor.org/packages/release/bioc/html/GenomicRanges.html (accessed on 4 May 2022)) and “*motifmatchr*” (https://bioconductor.org/packages/release/bioc/html/motifmatchr.html (accessed on 6 May 2022)). *GenomicRanges* allows analyzing the genome by dividing it into predefined intervals, but no information about the genomic feature is produced. The package *motifmatchr* finds motifs along the genome, but no gene or protein annotations are included in the output. Moreover, MeStudio simply takes as input for the motifs a text file, which is a more user-friendly format compared to the one required by *motifmatchr*. Table 1 provides a comparison of the features of MeStudio to tools for similar purposes.

### 2.2. The Sinorhizobium Case Study

In order to show the performance of MeStudio, the genomic sequences of two strains of the model symbiotic nitrogen-fixing bacterium *S. meliloti* were produced and analyzed together with two additional recently published SMRT data [12] for a total of four genomic sequences of *S. meliloti* strains, 2011, FSM-MA, BE31LL, and BO21CC (Appendix A). On the SMRT assembled reads of the genomes of the strains, MeStudio was able to identify a total of 26 motifs (Figure 1). All but six motifs (namely, CTYCCAG, DCTGCAGGS, GCCGGCYD, RAGCWGCTY, RCCAGCC, and RCTGCAGGS) were common to the four strains. The number of retrieved methylated sites ranged from a few units (especially for private motifs, those present in one strain only) to several thousands (such as GANTC, which is a classical motif methylated by the CcrM DNA methylase and involved in cell-cycle regulation [23]. CDS and nCDS showed similar frequencies (Figure 1) (Appendix A), as expected for methylation being present on both DNA strands. Intergenic sequences (tIG) showed the lowest number of methylated sites, while upstream sequences to a gene (US), *bona fide* corresponding to putative promoter regions, reported values generally one order of magnitude higher than tIG, and, in some cases, differences in values between strains ranged around twofold (e.g., CTYCCAG and GCCAGG). Furthermore, differences in the abundance of methylated profiles are evident if we consider the two strains grown until the late exponential phase in minimal medium (i.e., FSM-MA and 2011) and those grown in TY medium (i.e., BE31LL and BO21CC). Lastly, the presence of motifs in one strain only may suggest the occurrence of strain-specific restriction-modification systems, although the small number of methylated sites may also suggest alternative hypotheses (i.e., methylation on some genomic regions only related to regulation of expression at specific loci).

In conclusion, we encourage the use of MeStudio to unearth epigenomic data which are interpretable in a comparative genomic framework: the correlation among a methylation position, motifs of interest, and the protein annotation related to a CDS region strengthens the inference between the epigenetic modification and its functional role.

## 3. Materials and Methods

### 3.1. Bacterial Strains and Culture Conditions

Strains of *S. meliloti* BE31LL and BO21CC were resuscitated from glycerol stock tubes (codes BM932, BM936) stored at −80 °C in the collection of the Laboratory of Microbial Genetics, Dep. Of Biology, University of Florence, Italy. After re-isolation on TY medium agar plates [24] (tryptone 5 g/L, yeast extract 0.4 g/L, CaCl_2_ 0.4 g/L, and agar 7.5 g/L), single colonies were inoculated in 5 mL liquid TY medium and grown under constant agitation (125 rpm) at 30 °C.

### 3.2. DNA Extraction and SMRT Sequencing

DNA was extracted from overnight cultures (OD_600nm_ = 1.5) using PowerSoil DNA Isolation Kit (Qiagen, Hilden, Germany). After quantification by gel electrophoresis and fluorimetric assay (Qubit, Thermo Fisher Scientific, Waltham, MA, USA), we followed the procedure already reported in [25] for fragmenting DNA with g-TUBE (Covaris Inc., Woburn, MA, USA) to an average 15 kbp size and preparing the sequence library using the Pacific Biosciences SMRTbell Express Template Prep Kit 2.0 (Pacific Biosciences, Menlo Park, CA, USA). Sequencing was performed on a Sequel apparatus (Pacific Biosciences, Menlo Park, CA, USA) by SMRT technology [21], using Sequel Sequencing Kit 3.0.

### 3.3. Sequence Analysis and Annotation

The obtained SMRT reads were assembled using the SMRT Link software ver. 8.0.0.80529 (Pacific Biosciences, Menlo Park, CA, USA), producing oriC-oriented assemblies. Annotation was performed using Prokka v1.14.5 [26]. Sequences were deposited in the NCBI database and are available under accession numbers SAMN16976749 and SAMN16976751 (BioProject PRJNA681719). Two additional genomic sequences were analyzed corresponding to *S. meliloti* strains 1021 and FSM-MA, deposited under BioProject PRJNA705832 [12].

### 3.4. Software Design and Implementation

MeStudio consists of several tools that can be run individually or as part of a pipeline, and it uses a naïve string-matching algorithm to map motif sequences to the reference genome. The required input data consist of only three files: (i) a FASTA file containing the genome sequence, (ii) a genomic annotation file in GFF3 format, and (iii) another GFF3 containing the methylated nucleotide positions. The latter is automatically generated from the output of the SMRTlink software of Pacific Biosciences DNA sequencers. As a result, MeStudio produces several files including: (i) a text file with summarized statistics of the methylation occurrences along the genomic features, (ii) distribution and circular plots, and (iii) BED files containing protein annotation of the genes in which methylated motifs have been found. A complete workflow is provided in Figure 2. Demo files for input and outputs are available at https://github.com/combogenomics/MeStudio (accessed on 4 October 2022).

### 3.5. Preprocessing

In the first instance, MeStudio performs a quality check via a preprocessing Python script named *ms_replacR*. For a proper analysis, MeStudio needs consistent formatting on the sequence identifiers of the three main input files: the genomic annotation (GFF3 format), sequencer-produced modified base calls (GFF3 format), and the genomic sequence (FASTA file). Since these files may derive from different sources, it is possible for the user to experience differences in the syntax and/or annotation of the sequence identifiers (the “seqid” field); to avoid these possible inconsistencies, *ms_replacR* copies the original files into the output directory, performs a quality check, and corrects the errors, if needed. The most frequent formatting issue we have encountered is the presence of a pipe character and an underscore used interchangeably and inconsistently across files deriving from several sources. To correct for this possible incompatibility issue, by default, the pipe symbol is replaced by the underscore as a separator. More details are provided in the MeStudio manual on GitHub.

### 3.6. Core Processing

The processing of the input files is handled by five executables which we refer to as the “MeStudio Core”. These components match the nucleotide motifs to the genomic sequence and map them to the corresponding category, which are extracted from the annotation file. Categories are defined as follows: (i) protein-coding genes with an accordant (sense) strand (CDS), (ii) a discordant (antisense) strand (nCDS), (iii) regions that fall between annotated genes (true intergenic, tIG), and (iv) regions upstream of the reading frame of a gene, with an accordant strand (US) (Figure 2B). The CDS feature is defined by the ORF, and the nCDS is its corresponding on the antisense strand. The tIG is defined as the region between two different ORFs on both strands, as reported in Figure 2B. The US region is defined, by default, as the portion of the genome between the end of an ORF and the beginning of the next one; on the other hand, it is possible to set a personalized upstream range via an appropriate flag. The current implementation uses an optimized naive string-matching algorithm to map motif sequences to the reference genome. During the matching stage, each replicon or chromosome is loaded in memory, and both strands are scanned for the presence of the motif sequences, which can obviously hold ambiguous characters. Time complexity in the worst-case scenario is O(m × (n − m + 1)) + alpha, with alpha being an integer proportional to the number and realization of ambiguity characters present in each motif. All the motifs to be searched must be collected by the user and saved in an appropriate newline-delimited text file. The resulting binary files are then processed by another executable that is called for the task at hand. MeStudio Core crosses methylated base positions relative to the reference sequence starting with the previously described features, producing GFF3 files that serve as input for the final analysis stage. This is a computationally expensive part of the pipeline in which multiple nested for loops and calculations are performed. Integrating one motif on a four-contig genome (6,973,268 bp, 23,433 GANTC motif matches) took 0 min 27.116 s on a single AMD Opteron 6380 processor (2.5 GHz).

### 3.7. Postprocessing

MeStudio implements a postprocessing Python script named *ms_analyzR* which uses MeStudio Core results to produce analytical statistic outputs and return to the user graphical outputs and tables (in the form of BED files), which can be directly parsed using R. In addition, to strengthen the pipeline with comparative genomic analyses, the “gene_presence_abscence.csv” file produced by Roary [27] is needed to define the methylation level and patterns of core and dispensable genome fractions, as well as to annotate the genes-coded proteins. *ms_analyzR* logs the total number of genes found for each category (CDS, nCDS, tIG, and US). Additionally, methylation data are shown, such as (i) the total number of methylated sites, (ii) the total number of methylated genes, (iii) the ID of the most methylated gene (geneID), and (iv) the product of that gene. Integrating data from Roary is functional to characterize the geneID associated with the name of the protein (as annotated by Prokka [26]) as part of the core or dispensable genome. All the information is saved into a log file, together with plots accounting for the distribution of the methylations (Figure 3). To ensure customizability, *ms_analyzR* also includes two optional flags: “—make_chrom” and “—make_bed”. The “—make_chrom” flag saves into the previously specified output directory the GFFs at the “chromosome level” rather than the “feature level”. Each GFF produced is characterized not by feature (CDS, nCDS, tIG, and US) but by chromosomes (or contigs), maintaining the MeStudio Core-derived contents and layout. The “—make_bed” flag produces a BED file for each feature reporting (i) the *chrom* column, with the name of each chromosome or contig, (ii) the start and (iii) end of the feature, (iv) the name of the geneID found in that interval, (v) the number of methylations found for geneID, and (vi) the protein product of the ID. Information contained in BED files can be readily used to plot the distribution of the methylation density for each feature, making use of the *circlize* R package (https://github.com/jokergoo/circlize (accessed on 18 July 2022)) (Figure 4); an R script for this purpose is already available on our GitHub.

## 4. Conclusions

We reported here the description of a novel software called MeStudio, for the analysis of DNA methylation profiles obtained by single-molecule real-time sequencing. MeStudio has several novel and useful features compared to the few existing tools, as it provides outputs in the form of GFF and BED files which contain information on the position of methylated sites and methylated motifs, the number of methylated sites and profiles for each genomic feature, and graphical outputs, as well as protein annotation. The genomic features analyzed include genic and intergenic regions (comprising putative promoters), allowing the formulation of hypotheses related to the importance of DNA methylation on the regulation of gene expression and on other relevant biological phenomena [28]. In addition to being developed for prokaryotic genomes, MeStudio can handle any kind of sequence, by simply providing a suitable set of input files (Figure 2A). By providing information on motif occurrence and genomic localization, MeStudio contributes to the basis for comparative analyses of DNA methylation profiles among strains, in terms of evolutionary studies on populations and species, as well as epigenomic modifications during adaptation and development.

Lastly, MeStudio is very user-friendly given its easy installation and its possibility to be run as a pipeline in a single command line call. We developed the scripts in Mac OS and Linux kernel environments, with the possibility of expansion to Windows platforms. Moreover, we plan to make MeStudio affordable to ONT data.

## Figures and Tables

**Figure 1 ijms-24-00159-f001:**
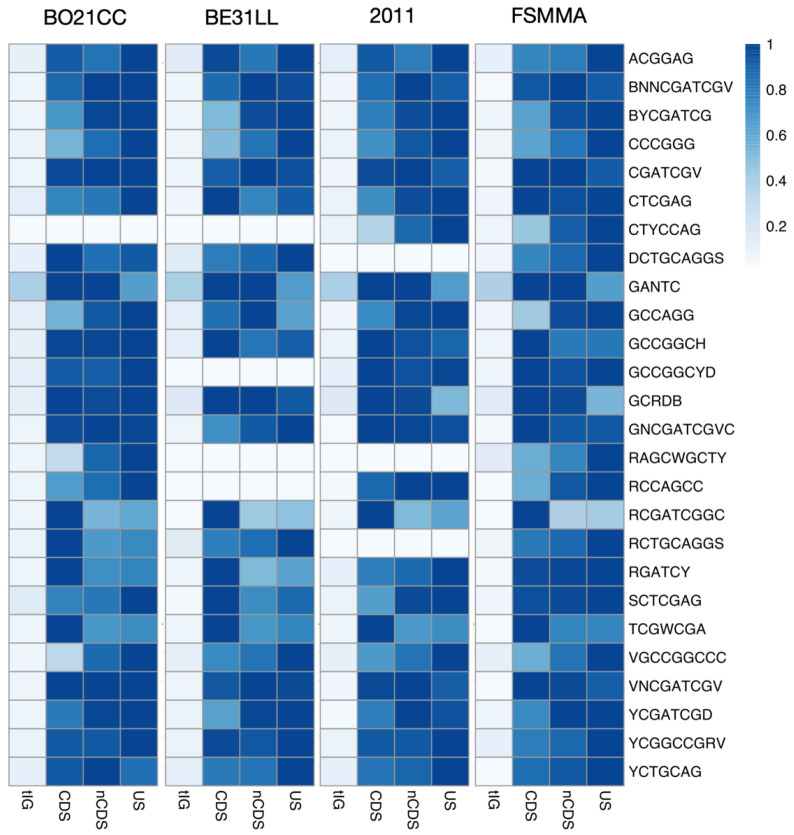
Heatmap representing methylated motif’s occurrences scaled as ratio to the max.

**Figure 2 ijms-24-00159-f002:**
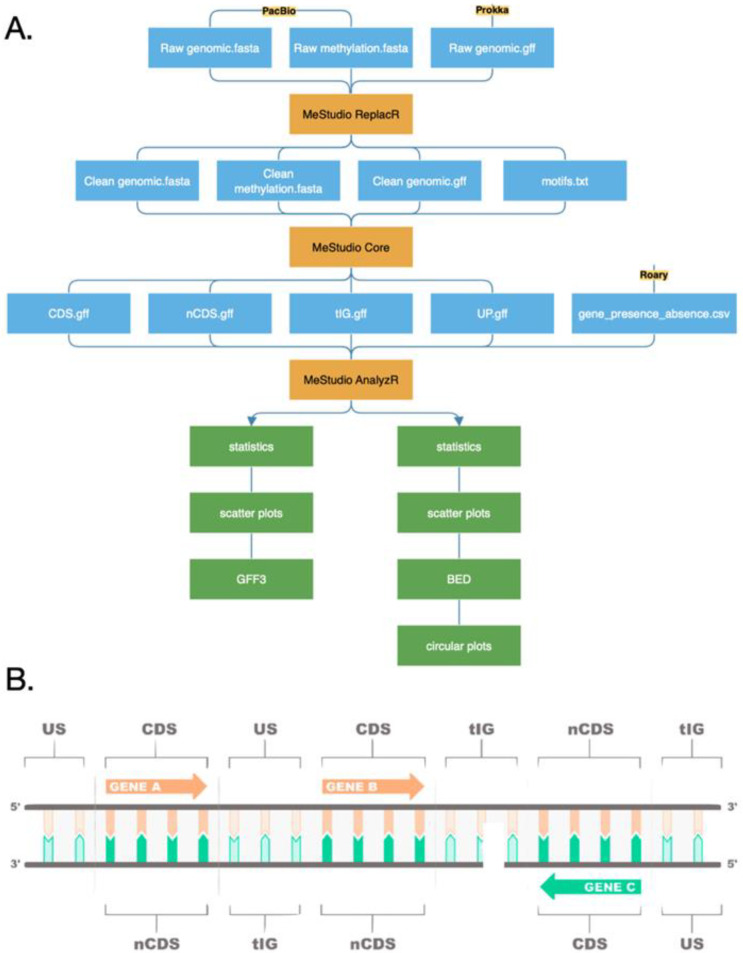
MeStudio overview. (**A**) Workflow. Each blue block represents input files. The orange blocks indicate the scripts. The green boxes indicate output files. (**B**) Graphical representation of the used terminology: CDS, coding sequence; nCDS, coding sequence opposite strand; tIG, intergenic sequence between two genes in opposite directions; US, upstream sequence to a coding sequence (intergenic sequence between two genes having the same orientation). See text for details.

**Figure 3 ijms-24-00159-f003:**
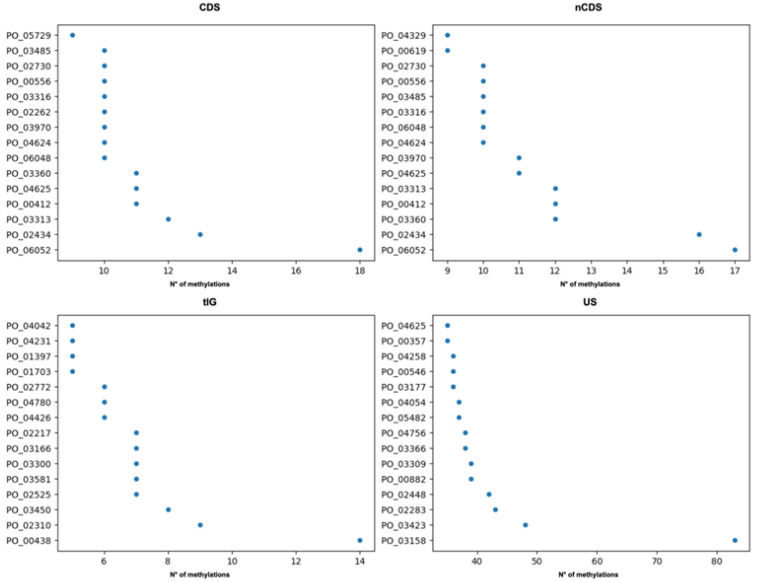
Scatter plots of GANTC motif in *S. meliloti* FSM-MA. The Y-axis reports geneIDs, whereas the X-axis reports the number of methylations found for each geneID. GeneIDs are taken from the annotation (see GitHub repository for the annotation files: https://github.com/combogenomics/MeStudio (accessed on 4 October 2022)). Plots for the different categories of methylated sites (CDS, nCDS, tIG, and US) are reported.

**Figure 4 ijms-24-00159-f004:**
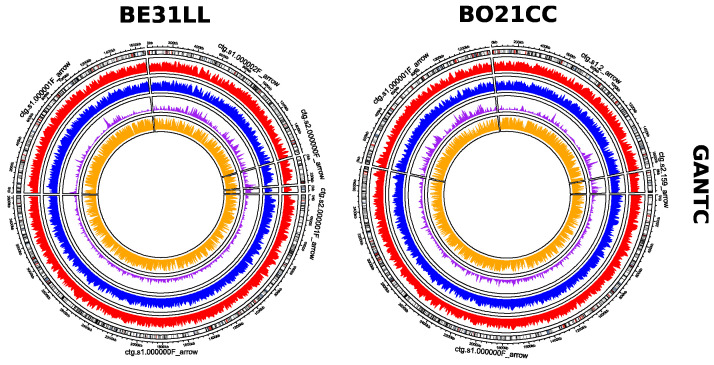
Circular density plots of GANTC and GCCCGGCH motifs in FSM-MA and 1021 strains of *S. meliloti*. The outer circle represents the genome annotation of the contigs of the strain (black lines indicate the position of CDS). Each inner circle represents a different category of methylated sites, CDS (red), nCDS (blue), tIG (purple), and US (yellow). The bars of each plot indicate the values for each category.

**Table 1 ijms-24-00159-t001:** MeStudio features compared to existing tools.

Tool	Programming Language	Motif Recognition	Motif Matching with Respect to Genomic Features	Graphical Outputs	Reference
MeStudio	Python, C	Yes	Yes	Yes	This study
*GenomicRanges*	R, C	No	No	Yes	Bioconductor package
*motifmatchr*	R, C++	Yes	Yes (only providing genomic ranges)	Yes	Bioconductor package
*Meta-epigenomics*	Python	Yes	No	No	https://github.com/hoonjeseoho/Meta-epigenomics (accessed on 19 June 2022)
Methplotlib	Python, Bash	No	No	Yes	De Coster et al. (2020) [21]
a-slide/pycoMeth	Python, Bash	No	No	Yes	Leger (2020) [20]
NanoMethViz	Python, Bash	No	No	Yes	Su et al. (2021) [19]

## Data Availability

The data that support the findings of this study are openly available on GitHub at https://github.com/combogenomics/MeStudio (accessed on 4 October 2022).

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
