# Peer review of "Crossing Bacterial Genomic Features and Methylation Patterns with MeStudio: An Epigenomic Analysis Tool"

_ijms, 2022, doi:10.3390/ijms24010159_

Round 1
Reviewer 1 Report
This ms presents as a new pipeline for the analysis of methylation profiles based on SMRT sequencing data and this is applied to some Sinorhizobium strains. This ms covers a current gap existing for the simple analysis of SMRT and methylome studies in prokaryotes. I have no major comments to this study although I would like or miss a few points that the authors could consider to include in the next version of the ms.
I miss some comments on the mechanisms of the software to carry out some steps. For instance, how is performed the matching to the genome or which algorithms are used or if it follows previous procedures/algorithms used in other programs,...
The pipeline according to the authors performs some statistics but I do not see clear examples on this in the ms. Which are the statistics that are calculated? In addition, Figures 2 are difficult to compare or extract comparatively some differences. I would suggest to include some comparisons that clearly reflect differences among the studied strains, for example, figures on different methylated motifs or frequency distribution of methylation motifs for the different strains, etc.
In Figure 1, I do not see the "viz." in the Figure 1B.
In Table 1, please, correct "Bioncoductor". This should be Bioconductor package.
Table 2 is hard to read and visualize. Is there another way to visualize this information? The whole table could be provided ans supplementary material and the ms could present the information with some figures.
Author Response
Rev. 1
This ms presents as a new pipeline for the analysis of methylation profiles based on SMRT sequencing data and this is applied to some Sinorhizobium strains. This ms covers a current gap existing for the simple analysis of SMRT and methylome studies in prokaryotes.
Q1. I have no major comments to this study although I would like or miss a few points that the authors could consider to include in the next version of the ms. I miss some comments on the mechanisms of the software to carry out some steps. For instance, how is performed the matching to the genome or which algorithms are used or if it follows previous procedures/algorithms used in other programs, ...].
Reply: We appreciate Rev. 1's suggestion to include additional details relative to the mechanisms of the software. The optimized naive string matching is a well-known algorithm for pattern matching when patterns are few and relatively small. The choice of the algorithm works under the reasonable assumption that, in our experience, the amount of input meaningful motifs seldom even exceeds a dozen. In compliance to Rev. 1's recommendations we have updated the paragraph at line now 286, integrating standard algorithm information, i.e., name and time complexity. The sentences now read as follows: "The current implementation uses an optimized naive string-matching algorithm to map motif sequences to the reference genome. During the matching stage, each replicon or chromosome is loaded in memory and both strands are scanned for the presence of the motif sequences, which can obviously hold ambiguity characters. Time complexity in the worst-case scenario is O(m*(n-m+1)) + alpha, with alpha being an integer proportional to the number and realization of ambiguity characters present in each motif. All the motifs to be searched must be collected by the user and saved in an appropriate newline-delimited text file."
Q2. The pipeline according to the authors performs some statistics but I do not see clear examples on this in the ms. Which are the statistics that are calculated?
Reply: We acknowledge that the used terminology was unclear. Indeed, with the term "statistical analyses" we really meant summarized descriptive statistics. Therefore, in conformity with the objection that Rev. 1 judiciously raised, we've updated the paragraph starting at line now 302 which now reads:"MeStudio implements a post-processing Python script named ms_analyzR which uses MeStudio Core results to produce analytical statistic outputs and return to the user graphical outputs and tables (in the form of BED files) which can be directly parsed using R."
Q3. In addition, Figures 2 are difficult to compare or extract comparatively some differences. I would suggest to include some comparisons that clearly reflect differences among the studied strains, for example, figures on different methylated motifs or frequency distribution of methylation motifs for the different strains, etc.
Reply: We thank Rev. 1 for observing the need of a qualitative comparison among the strains to unearth differences between the motif’s frequency distributions taking into account the genomic compartmentalization. Circular plots are popular representation that provide holistic visualization of high throughput large scale data. However, we perfectly agree that they are difficult to read to extract clear differences. For this reason, MeStudio outputs include numerical data (as for instance frequency distribution of methylation motifs) which allow to extract clear differences among studied strains. A graphical representation of such outputs is now reported in Figure 1, where methylated motif’s occurrences are shown on a heatmap (see reply to Q6). However, to better appreciate distribution of the methylated motifs along the genome with respect to the genomic features (CDS, nCDS, tIG and US) we have now included this information on a separate Figure with respect to the scatter plot visualization of the number of methylations found for each geneID. Consequently, Figures 2a and 2b are now respectively split into Figure 2 and Figure 3 for a more pleasurable examination.
Q4. In Figure 1, I do not see the "viz." in the Figure 1B.]
Reply: Reviewed.
Q5. In Table 1, please, correct "Bioncoductor". This should be Bioconductor package.
Reply: Reviewed.
Q6. Table 2 is hard to read and visualize. Is there another way to visualize this information? The whole table could be provided as supplementary material and the ms could present the information with some figures.
Reply: We agree with Rev. 1’s suitable observation. Table 2 is now disposed as supplementary material and replaced by Figure 1 in which numerical values have been normalized with ratio to the maximum and represented through stacked columns histogram.
Reviewer 2 Report
In the submitted manuscript, the authors developed a new bioinformatics pipeline named MeStudio for epigenomic analysis. Several similar pipelines are available online for different sequencing platforms, but the authors implemented some unique features in this pipeline and showed a case study in this manuscirpt. Overall, this manuscript is easy to follow, ideas and critical points of this manuscript are clear. Additionally, instructions in GitHub are straightforward. However, the authors should provide some crucial information in the introduction, method, and results/discussion sections. I list my comments here for the authors ’consideration.
1. In the introduction section, the authors need to provide more details regarding similar pipelines for different sequencing platforms and point out the limitations of currently available tools.
2. In the method section, the authors should have mentioned the description of MeStudio, such as modules, algorithms, or approaches to design your pipeline. I did find the information in your GitHub, and you can transfer it into this manuscript.
3. In the results/discussion section, the authors need to do some benchmarks to demonstrate the performance of your tool is better than others for accuracy, speed, etc., using at least one publicly available dataset.
Author Response
Rev. 2
In the submitted manuscript, the authors developed a new bioinformatics pipeline named MeStudio for epigenomic analysis. Several similar pipelines are available online for different sequencing platforms, but the authors implemented some unique features in this pipeline and showed a case study in this manuscript. Overall, this manuscript is easy to follow, ideas and critical points of this manuscript are clear. Additionally, instructions in GitHub are straightforward. However, the authors should provide some crucial information in the introduction, method, and results/discussion sections. I list my comments here for the authors ’consideration.
Q1. In the introduction section, the authors need to provide more details regarding similar pipelines for different sequencing platforms and point out the limitations of currently available tools.
Reply: We thank Rev. 2 for highlighting the importance of the existence of similar pipelines for different sequencing platforms. We have indeed already included a body of references to these tools in the introduction, and with all due respect, further expanding on third party software for a completely different sequencing technology would offset the main theme of our paper, which is to introduce an unprecedented application for crossing data in SMRT context.
Q2. In the method section, the authors should have mentioned the description of MeStudio, such as modules, algorithms, or approaches to design your pipeline. I did find the information in your GitHub, and you can transfer it into this manuscript.
Reply: We agree with Rev. 2 that the description of the technical aspects of MeStudio should be included in the methods section. We have therefore transferred much of the paragraphs residing in the Results and Discussion, to the Materials and Methods, as thoughtfully suggested by Rev. 2.
Q3. In the results/discussion section, the authors need to do some benchmarks to demonstrate the performance of your tool is better than others for accuracy, speed, etc., using at least one publicly available dataset.
Reply: The relatively recent introduction of SMRT sequencing potential to detect methylated bases, unfortunately, doesn't allow a fair and thorough comparison to any of the existing packages which can only perform some of the many steps needed for even a basic analysis. This can be found in paragraph 2.1 Tool-wide Comparison in our manuscript. Needless to say, we're excited to bring novelty in the field of molecular sciences, and to describe the hereby presented data using software that will exist in an open-source framework.
Round 2
Reviewer 2 Report
The authors have made changes according to the previous review.